# Mineral-like Synthetic Compounds Stabilized under Hydrothermal Conditions: X-ray Diffraction Study and Comparative Crystal Chemistry

**Galina Kiriukhina** [1,2], **Olga Yakubovich** [1,*], **Polina Verchenko** [1], **Anatoly Volkov** [3], **Larisa Shvanskaya** [1], **Olga Dimitrova** [1] and **Sergey Simonov** [4]

1   Department of Crystallography, Geological Faculty, Lomonosov Moscow State University, Leninskie Gory 1, 119991 Moscow, Russia; g-biralo@yandex.ru (G.K.); yapoletta@mail.ru (P.V.); lshvanskaya@mail.ru (L.S.); dimitrova@list.ru (O.D.)
2   Korzhinskii Institute of Experimental Mineralogy of Russian Academy of Sciences, 142432 Chernogolovka, Russia
3   Skolkovo Institute of Science and Technology, 121205 Moscow, Russia; toljha@yandex.ru
4   Institute of Solid State Physics of Russian Academy of Sciences, 142432 Chernogolovka, Russia; simonov@issp.ac.ru
*   Correspondence: yakubovich.olga320@gmail.com; Tel.: +7-903-975-91-06

**Abstract:** Under hydrothermal conditions emulating natural hydrothermalites, three oxo-salts with sodium and transition metal cations were obtained in the form of single crystals. Their compositions and crystal structures were studied using scanning electron microscopy, microprobe X-ray spectral analysis, and X-ray single-crystal diffraction. The sodium cobalt silicate, i.e., $Na_2CoSiO_4$, a structural analog of the mineral liberite, is well known as an ionic conductor. Its crystal structure consists of a framework derived from $\beta$-tridymite, built using the Co- and Si-centered tetrahedra sharing vertices. The sodium oxocuprate phosphate chloride $Na_2Cu_3O(Cu_{0.8}Na_{0.2})(PO_4)_2Cl$ belongs to a group of compounds, including fumarolic minerals, characterized by the presence of oxo-centered pyroxene-like chains in their structures. The crystal structure of mineralogically probable sodium vanadium phosphate hydroxide ($Na_3V(OH)(HPO_4)(PO_4)$) is based on chains built using octahedra centered by magnetically active $V^{3+}$. Magnetic susceptibility measurements indicate an antiferromagnetic arrangement of $V^{3+}$ ions and no transition to an ordered state up to 2 K.

**Keywords:** sodium and transition metal oxo-salts; silicate; phosphate; synthetic analogs of minerals; liberite; single-crystal X-ray diffraction; hydrothermal synthesis; oxo-centered chains; antiferromagnetic properties; ionic conductivity



## 1. Introduction

Nature has long served as a source of inspiration for scientists. Minerals, renowned for their geological stability, have played a significant role in the chemistry of materials. However, it is a well-known fact, both among materials scientists and mineralogists, that natural compounds often contain a multitude of impurities. These impurities frequently interfere with the study of the physical properties of minerals, hindering their industrial applications. To solve this problem, the laboratory gradually adopted an experimental approach that mimics the natural conditions of mineral formation.

The natural pegmatite and hydrothermal process occur in superheated water under pressure beneath the earth's surface in a temperature range of 100–700 °C. The hydrothermal technology takes its name from hydrothermal vents where extreme temperatures and pressures occur. A distinctive feature of hydrothermal vents is the presence of supercritical water (374 °C and a pressure of 218 atm), with its unique properties to dissolve solids like a liquid yet flow with next to no viscosity like a gas. By working at the upper end of the temperature range (around 500–700 °C), it is possible to achieve conditions roughly similar

to those found in natural pegmatite systems. Even under these temperature (and pressure) conditions, the liquid remains sufficiently polar to partially dissolve most metal oxyanions. Synthesis at intermediate temperatures of 100 °C < T < 374 °C allows it to remain well below the critical point of water but still heat the vessel above the normal boiling point. By keeping the reaction closed, water can be maintained in a liquid state well above the point at which it would normally boil. As long as the critical point is avoided, the pressures required for this approach are much more controllable. Undoubtedly, both thermal regimes are characteristic of the geological system [1].

These temperatures and chemical environments inherent in the hydrothermal process can be simulated in the laboratory. Therefore, by exploiting the similarities between mineral forming conditions and laboratory techniques, it becomes possible to investigate the structural relationships with complex structures derived from hydrothermal fluids, using the natural materials as a motivation. This approach allows for the creation of chemically "pure" analogs of minerals from various chemical classes, including rock-forming oxides and silicates, as well as their recrystallized products. Thus, silicates of rare-earth metals, structurally related to minerals such as wadeite, fresnoite, apatite, olivine, and garnet, were obtained using high-temperature hydrothermal techniques. The synthetic analogues have found applications as phosphors, lasers, scintillators, and Faraday rotators due to their optical properties [2]. In our research into crystallization processes at medium temperatures, we successfully produced structural analogs of minerals such as mahnertite ($NaCu_3(AsO_4)_2Cl \cdot 5H_2O$) [3] and ellenbergerite ($Mg_6(Mg,Ti,Zr)_2(Al,Mg)_6Si_8O_{28}(OH)_{10}$) [4]. These synthetic phases exhibit unique magnetic properties, particularly low-dimensional magnetism arising from the presence of chains or layers of octahedra centered by open-shell metal atoms in their structures. Past reviews of synthetic mineral analogs [5,6] highlighted a broad range of compounds with ion-conducting properties. The synthetic equivalents of minerals such as triphylite ($LiFePO_4$), niahite ($(NH_4)(Mn,Mg)(PO_4) \cdot H_2O$), simferite ($LiMg(PO_4)$), tavorite ($LiFe(PO_4)(OH)$), marićite ($NaFe(PO_4)$), sarcopside (($Fe,Mn,Mg)_3(PO_4)_2$), and alluaudite (($Na,Ca)Mn(Fe,Mn,Fe,Mg)_2(PO_4)_3$) are actively utilized as electrode materials for sodium- or lithium-ion batteries.

Evidently, there is a substantial interest in the field of materials science not only in direct mineral analogs but also in determining their structural derivatives. By modifying the natural conditions of mineral formation in laboratory experiments by adjusting factors such as chemical reagents, temperature, pressure, and mineralizers, a wide array of mineralogically plausible compounds can be obtained. In this paper, we present the experimental results of crystallization in hydrothermal systems that mimic the expected hydrothermal conditions. We successfully synthesized three sodium-containing compounds: $Na_2CoSiO_4$ (I), a structural analog of mineral liberite ($Li_2BeSiO_4$) and the mineralogically reasonable phosphates $Na_2Cu_3O(Cu_{0.8}Na_{0.2})(PO_4)_2Cl$ (II) and $Na_3V(OH)(HPO_4)(PO_4)$ (III). All investigated phases were previously obtained in the form of polycrystalline samples, and their crystal structures were studied using powder X-ray diffraction. The synthesis of these compounds in the form of single crystals allowed us to obtain more precise structural data, localize hydrogen atoms, refine their positions, and analyze hydrogen bonds. Moreover, studies have been carried out on the magnetic properties of $V^{3+}$-containing phosphate.

## 2. Materials and Methods

### 2.1. Hydrothermal Synthesis

Single crystals were obtained in three different hydrothermal phosphate and silicate systems with cations of alkali and transition metals (Table 1). The initial chemically pure reagents were weighed, thoroughly ground in an agate mortar, and placed into an autoclave. In the silicate system, sodium hydroxide was added to improve the solubility of $SiO_2$, raising the solution's pH to 12. In the copper phosphate system, a small amount of hydrochloric acid solution was used to achieve an acidic medium with a pH of 1.5. In the phosphate system with vanadium, a small amount of lithium carbonate served as a mineralizer. Different temperature regimes were used for synthesizing these single crystals. For

270 °C middle-temperature hydrothermal conditions, the routine steel autoclave lined with fluoroplastic was utilized. However, 450 °C high-temperature hydrothermal experiments were conducted using a copper-coated high-pressure vessel of our own production made from a nickel–chromium alloy. It is critical for safety reasons to use superalloy autoclaves, as the creep rupture stress significantly enlarges with the increase in temperature. Distilled water was added according to the selected percentage of autoclave volume filling required to achieve the desired internal pressure. All experiments lasted for 10 days, after which period the autoclaves were left to cool naturally for 24 h. Crystallization products were washed multiple times with warm distilled water and air-dried. All three phases are shown in Table 1, where they are denoted as (I), (II), and (III), respectively.

**Table 1.** Secondary-electron SEM images showing the sample morphologies and experimental conditions of hydrothermal synthesis.

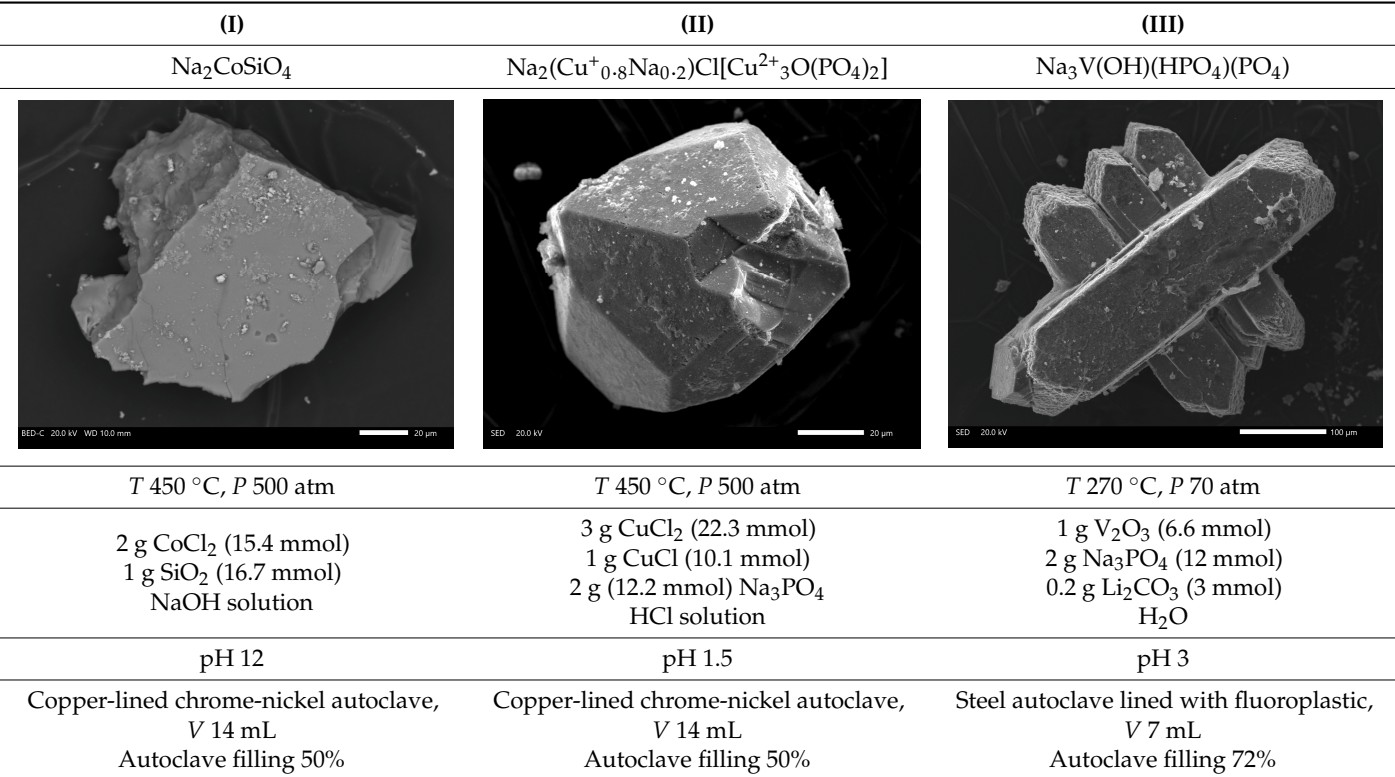

| (I) | (II) | (III) |
|---|---|---|
| $Na_2CoSiO_4$ | $Na_2(Cu^+_{0.8}Na_{0.2})Cl[Cu^{2+}_3O(PO_4)_2]$ | $Na_3V(OH)(HPO_4)(PO_4)$ |
| $T$ 450 °C, $P$ 500 atm | $T$ 450 °C, $P$ 500 atm | $T$ 270 °C, $P$ 70 atm |
| 2 g $CoCl_2$ (15.4 mmol)<br>1 g $SiO_2$ (16.7 mmol)<br>NaOH solution | 3 g $CuCl_2$ (22.3 mmol)<br>1 g CuCl (10.1 mmol)<br>2 g (12.2 mmol) $Na_3PO_4$<br>HCl solution | 1 g $V_2O_3$ (6.6 mmol)<br>2 g $Na_3PO_4$ (12 mmol)<br>0.2 g $Li_2CO_3$ (3 mmol)<br>$H_2O$ |
| pH 12 | pH 1.5 | pH 3 |
| Copper-lined chrome-nickel autoclave, $V$ 14 mL<br>Autoclave filling 50% | Copper-lined chrome-nickel autoclave, $V$ 14 mL<br>Autoclave filling 50% | Steel autoclave lined with fluoroplastic, $V$ 7 mL<br>Autoclave filling 72% |

### 2.2. X-ray Spectral Analysis

High-quality crystals selected via light microscopy were subjected to X-ray spectral analysis. Semi-quantitative ED analysis and crystal images were obtained from unpolished samples through graphite sputtering, using an accelerating voltage of 20 kV and a current rate of 10 nA, with a spectrum accumulation time of 50 s. These measurements were conducted at the Laboratory of Analytical Techniques of High Spatial Resolution, the Department of Petrology, the Faculty of Geology, Lomonosov Moscow State University, utilizing a JEOL JSM-6480LV scanning electron microscope with EDS-spectrometer. As reference standards, stoichiometric compounds and natural minerals were used, such as metallic Co and Cu, NaCl for Na and Cl, $V_2O_3$ for V, GaP for P, and natural garnet and diopside for Si and O, respectively.

### 2.3. Single-Crytsal X-ray Diffraction

The obtained crystals were investigated using X-ray diffraction at low temperatures for (I) and (II) and in standard conditions for (III), using Mo$K\alpha$ radiation with a single-crystal diffractometer equipped with a CCD detector (Atlas-S2 & Sapphire3 detectors,

Rigaku Oxford Diffraction, Tokyo, Japan). The datasets were corrected for background, the Lorentz and polarization effects, and absorption [7]. Most calculations for structural studies were performed using the WinGX program system [8]. The crystal structures were solved via direct methods and refined against the $F^2$ data using the SHELX programs [9,10]. Structural data were deposited via the joint CCDC/FIZ Karlsruhe deposition service under the deposition numbers 2310942 **(I)**, 2310943 **(II)**, and 2310944 **(III)**. Cif-data can be obtained free of charge from FIZ Karlsruhe via the following webpage: www.ccdc.cam.ac.uk/structures, accessed on 29 November 2023. Also, cif and checkcif files can be found in the Supplementary Materials to this paper.

## 3. Results and Discussion

### 3.1. Chemical Composition

The X-ray spectral analysis of blue lamellar crystals **(I)** revealed the presence of Na, Co, Si, and O atoms; the brown isometric well-faced crystals **(II)** contained Na, Cu, P, Cl, and O in a Na:Cu:P:Cl ratio of 2.2:3.5:2:1. The gray-green prismatic crystals in intergrowths of **(III)** contained Na, V, P, and O atoms with a Na:V:P ratio equal to 3:1:2.

### 3.2. Crystal Structure Solution

Crystal data and details of data collection and refinement are presented in Table 2. The pseudo-orthorhombic **(I)** $Na_2CoSiO_4$ was refined as pseudo-merohedral microtwin with a ratio of twin components equal to 0.32:0.68 for $R_1 = 0.025$. During the solution of the crystal structure **(II)**, three symmetrically independent positions of copper atoms were recognized. The coordination geometry for one of these positions (Cu3) indicated the presence of monovalent copper, while the other two positions (Cu1, Cu2) exhibited distorted Jahn–Teller polyhedra typical of $Cu^{2+}$. The refinement revealed that the $Cu3^+$ site is "diluted" by sodium atoms at a ratio of 0.8:0.2. The refinement resulted in the crystal–chemical formula $Na_2(Cu^+_{0.8}Na_{0.2})Cl[Cu^{2+}_3O(PO_4)_2]$ for **(II)**. Data collection for the monoclinic phase **(III)** revealed non-merohedral twinning, connected by a pseudo-2-fold axis. A twin ratio between two components was refined at a ratio of 0.49:0.51. Two hydrogen atoms corresponding to two symmetrically independent OH groups were located via the electron density difference syntheses and refined based on the isotropic approximation. The final formula established through structural refinement ($R_1 = 0.043$) was $Na_3V(OH)(HPO_4)(PO_4)$. The formulae refined during the X-ray diffraction structural study of the compounds were consistent with the results of the microprobe analysis.

**Table 2.** The crystal data and experimental details of X-ray structural studies.

| | **(I)** | **(II)** | **(III)** |
|---|---|---|---|
| **Structural Formula** | $Na_2CoSiO_4$ | $Na_2Cu_3O(Cu_{0.8}Na_{0.2})(PO_4)_2Cl$ | $Na_3V(OH)(HPO_4)(PO_4)$ |
| $M_r$ | 197.00 | 531.00 | 327.87 |
| Space group, Z | *Pn*, 2 | *Cmcm*, 4 | *C2/m*, 4 |
| Temperature (K) | 170 | 150 | 293 |
| Unit cell parameters *a, b, c* (Å) | 5.2271(5) | 13.6243(2) | 15.4157(10) |
| | 5.4198(4) | 10.3531(2) | 7.3107(4) |
| | 7.0466(6) | 6.3586(1) | 7.0556(4) |
| β (°) | 90.011(7) | 90 | 96.702(6) |
| V (Å³) | 199.63(3) | 896.90(3) | 789.73 (8) |
| Radiation | Mo *Kα* | Mo *Kα* | Mo *Kα* |

**Table 2.** *Cont.*

| Structural Formula | (I) Na$_2$CoSiO$_4$ | (II) Na$_2$Cu$_3$O(Cu$_{0.8}$Na$_{0.2}$)(PO$_4$)$_2$Cl | (III) Na$_3$V(OH)(HPO$_4$)(PO$_4$) |
|---|---|---|---|
| $\mu$ (mm$^{-1}$) | 4.69 | 9.56 | 1.86 |
| Crystal size (mm) | 0.15 × 0.06 × 0.02 | 0.17 × 0.11 × 0.07 | 0.19 × 0.09 × 0.06 |
| Diffractometer | Oxford Diffraction Gemini | | Xcalibur Sapphire3 |
| Number of reflections: measured independently based on [$I > 2\sigma(I)$] | 1800, 909, 894 | 8798, 732, 727 | 1352, 1352, 1043 |
| $R_{int}$ | 0.016 | 0.016 | - |
| $(\sin \theta / \lambda)_{max}$ (Å$^{-1}$) | 0.703 | 0.703 | 0.594 |
| $R$ [$F^2 > 2\sigma(F^2)$], $wR(F^2)$, $S$ | 0.025, 0.064, 1.03 | 0.018, 0.046, 1.29 | 0.043, 0.109, 1.03 |
| Number of refined parameters | 74 | 57 | 91 |
| $\Delta\rho_{max}$, $\Delta\rho_{min}$ (e Å$^{-3}$) | 0.46, −0.45 | 0.74, −0.91 | 0.64, −0.57 |
| Flack parameter * | 0.01(3) * | - | - |

\* The algorithm from [11] was used to determine the Flack parameter *x*.

### 3.3. Na$_2$CoSiO$_4$, a Structural Analog of the Mineral Liberite

#### 3.3.1. The Structural Description and Analysis of Interatomic Distances

In the crystal structure of Na$_2$CoSiO$_4$, the cobalt and silicon tetrahedra share all vertices, forming the tetrahedral anionic framework [CoSiO$_4$]$^{2-}$, derived from the structure of β-tridymite, as illustrated in Figure 1a. The framework is negatively charged and balanced by sodium cations located within the channels. Within the *bc* plane, we can observe cellular layers consisting of six-membered rings composed of SiO$_4$ and CoO$_4$ tetrahedra alternating with each other (Figure 1b). Unlike the crystal structure of β-tridymite, these layers exhibit a uniform orientation of tetrahedra, described as *UUUUUU* (where '*U*' signifies the upward orientation of the tetrahedra). These layers are interconnected along the *a*-axis through shared polyhedra vertices, maintaining the overall polarity of the structure.

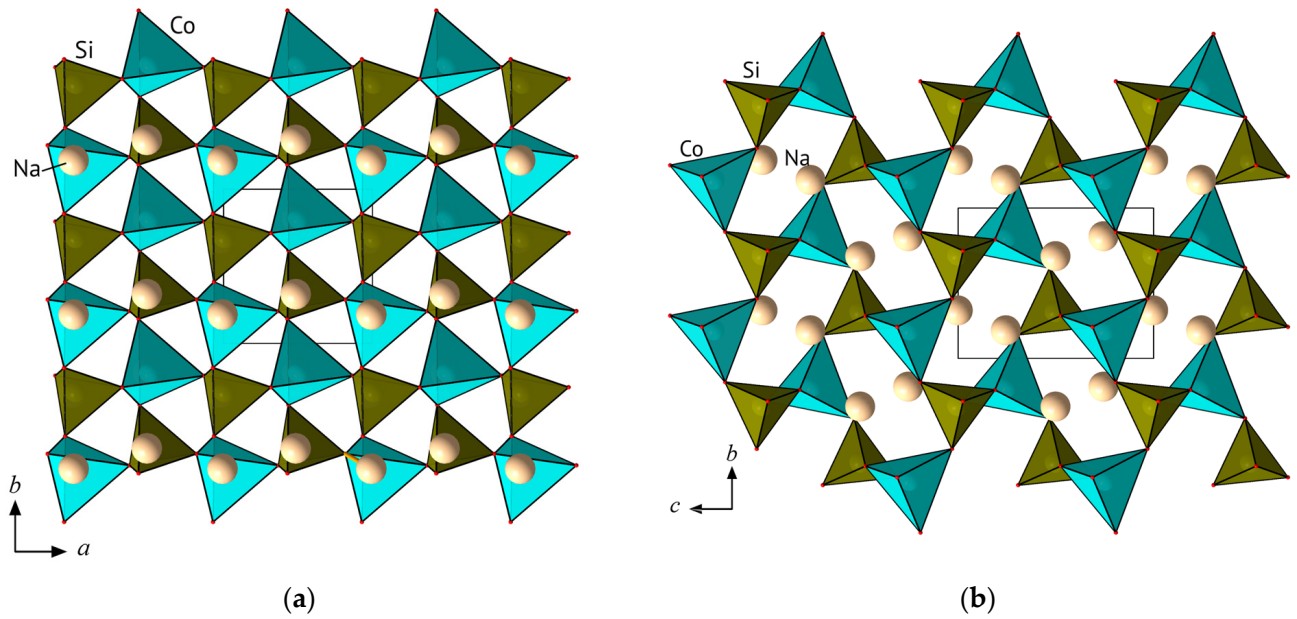

**Figure 1.** The projections of the Na$_2$CoSiO$_4$ crystal structure on the *ab* (**a**) and *bc* (**b**) planes.

In this monoclinic acentric crystal structure of $Na_2CoSiO_4$, all the atoms occupy general 2*a* Wyckoff positions. All Na, Co, and Si atoms are coordinated by oxygen ligands, forming tetrahedra (Figure 2). The Si–O bond lengths within the $SiO_4$ tetrahedron range from 1.618(8) to 1.644(8) Å, with an average value of 1.63 Å (Table 3). The larger $CoO_4$ tetrahedron exhibits Co–O distances in the interval 1.928(6)–1.969(5) Å (average 1.95 Å).

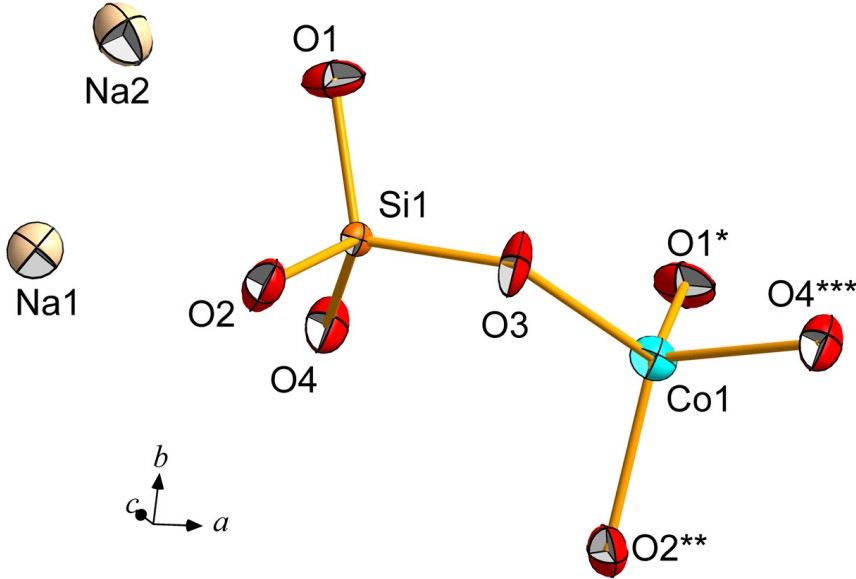

**Figure 2.** A symmetrically independent fragment of the crystal structure of $Na_2CoSiO_4$ **(I)**; thermal ellipsoids are shown with 90% probability. Symmetry operations: (*) 0.5 + *x*, 1 − *y*, and −0.5 + *z*; (**) 0.5 + *x*, −*y*, and −0.5 + *z*; (***) 1 + *x*, *y*, and *z*.

**Table 3.** Characteristic distances, Å, in the $Na_2CoSiO_4$ crystal structure.

| Co–Tetrahedron | | Si–Tetrahedron | |
|---|---|---|---|
| Co–O2 | 1.928(6) | Si1–O4 | 1.618(8) |
| –O3 | 1.944(5) | –O3 | 1.627(4) |
| –O1 | 1.958(4) | –O1 | 1.631(7) |
| –O4 | 1.969(5) | –O2 | 1.644(8) |
| <Co–O> | 1.95 | <Si–O> | 1.63 |
| **Na1–Tetrahedron** | | **Na2–Tetrahedron** | |
| Na1–O2 | 2.245(10) | Na2–O2 | 2.325(6) |
| –O3 | 2.272(10) | –O4 | 2.344(12) |
| –O4 | 2.279(5) | –O1 | 2.364(11) |
| –O1 | 2.331(10) | –O3 | 2.407(10) |
| <Na1–O> | 2.28 | <Na2–O> | 2.36 |

Sodium atoms occupy two symmetrically independent positions, both surrounded by four O ligands to form tetrahedra with Na1–O distances varying between 2.245(10) and 2.331(10) Å (average 2.28 Å). Meanwhile, the Na2–O distances are larger, ranging from 2.325(6) to 2.407(10) Å (average 2.36 Å). The next-to-nearest oxygen atoms associated with the Na1 and Na2 cations are situated at distances of 2.984(10) Å and 2.908(10) Å, respectively. Bond valence calculations show the impact of these next-to-nearest oxygens on the total valence of the sodium cations (Table 4).



**Table 4.** Na$_2$CoSiO$_4$. Bond valence data *.

|     | Na1           | Na2          | Co    | Si    | Σ    |
|-----|---------------|--------------|-------|-------|------|
| O1  | 0.203, 0.052  | 0.189        | 0.487 | 0.981 | 1.91 |
| O2  | 0.242         | 0.205        | 0.528 | 0.947 | 1.92 |
| O3  | 0.229         | 0.173, 0.061 | 0.506 | 0.992 | 1.96 |
| O4  | 0.226         | 0.197        | 0.473 | 1.016 | 1.91 |
| Σ   | 0.95          | 0.83         | 1.99  | 3.94  |      |

* The algorithm and empirical parameters from [12] were used.

Interestingly, the NaO$_4$ polyhedra form a cationic framework that exhibits topological similarity to the anionic framework built from Si- and Co-centered tetrahedra (Figures 1b and 3a). The three-periodic vertex-sharing of NaO$_4$ tetrahedra and interconnected CoO$_4$/PO$_4$ tetrahedra leads to quasi-layering in (101) projection. The [CoSiO$_4$] quasi-layers alternate in the [$\bar{1}$01] direction with topologically equal quasi-layers of Na-centered polyhedra (Figure 3b). The electrochemical properties of Na$_2$CoSiO$_4$, which will be discussed below, are associated with the migration of Na$^+$ ions through these quasi-layers parallel to the ($\bar{1}$01) planes (Figure 3).

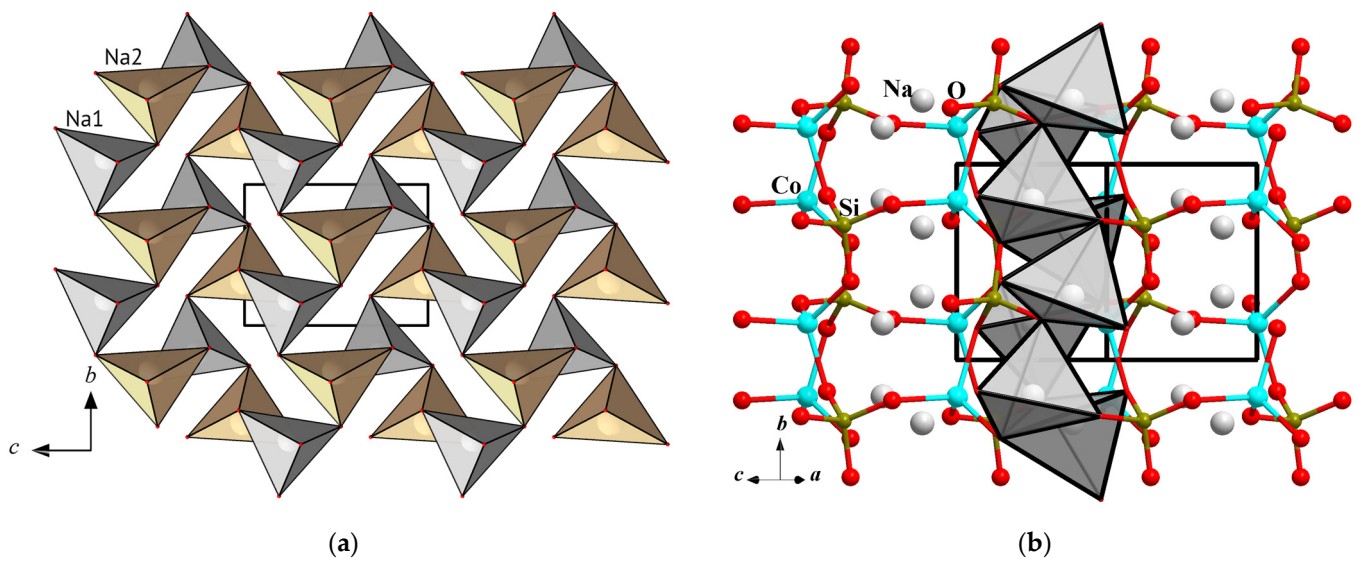

(**a**)                                                                       (**b**)

**Figure 3.** (**a**) The quasi-layer of sharing O vertices' NaO$_4$ tetrahedra in the Na$_2$CoSiO$_4$ crystal structure and (**b**) their alteration along [$\bar{1}$01] with topologically similar quasi-layers designed for Co- and Si-centered tetrahedra.

### 3.3.2. Na$_2$CoSiO$_4$ and the Liberite Structural Family of Ionic Conductors

The studied silicate, namely Na$_2$CoSiO$_4$, is a full structural analog of the mineral liberite, namely Li$_2$BeSiO$_4$, discovered in southern China [13]. Both liberite and disodium cobalt silicate crystallize in monoclinic or orthorhombic modifications, and both structures exhibit tetrahedral frameworks. Remarkably, the monoclinic polymorph of Na$_2$CoSiO$_4$ is entirely ordered, unlike the orthorhombic polymorph. In the orthorhombic structure Co and Si share the same positions, though the number of Na positions twice increased, and each site is a half populated.

Liberite and Na$_2$CoSiO$_4$ belong to a large group of compounds with the general formula $A_2MeXO_4$, where $A^+$ represents the alkaline cations Li or Na; $Me^{2+}$ represents Zn, Mn, Co, Mg, and Be; and $X^{4+}$ represents the semimetals Si or Ge. The crystal structures of $A_2MeXO_4$ phases are based on a distorted hexagonal close-packing of oxygen atoms, with half of the tetrahedral voids occupied by the $A$, $Me$, and $X$ atoms. The rich crystal chemistry of this family results from variations in the occupancy of the tetrahedral voids and the bonding of $A^+$-, $Me^{2+}$-, and $X^{4+}$-centered tetrahedra. Accordingly, the $A_2MeXO_4$

compounds crystallize in several polymorphic modifications, mainly in the monoclinic and orthorhombic space groups $P2_1$, $P2_1/n$, $Pn$, $Pmnb$, $Pmn2_1$, and $Pbn2_1$ [5]. The good $Na^+/Li^+$ ionic conductivity of the $A_2MeXO_4$ family arises from its structural peculiarities. Channels within the anionic tetrahedral framework allow for the migration of alkali metal cations. In these compounds, the $Na^+/Li^+$ deintercalation is compensated by two electrons per $Me^{2+}$ transition metal [14]. Moreover, the presence of transition metal atoms determines the magnetic properties of these phases due to the super-exchange magnetic interactions between $Me^{2+}$ cations through the $[XO_4]^{4-}$ anionic tetrahedra [15,16].

There was no clarity in the space group and unit cell parameters of the monoclinic polymorph of $Na_2CoSiO_4$, since earlier structural studies were conducted using polycrystalline $Na_2CoSiO_4$ samples synthesized through solid-state reactions [17,18]. These difficulties were apparently connected to the $Na_2CoSiO_4$ pseudo-orthorhombic symmetry of actually monoclinic crystals with $\beta$ close to $90°$, which exhibited a pseudo-merohedral twinning. Recently, by means of high-resolution X-ray powder diffraction using a synchrotron radiation source, a group of researchers [19] demonstrated the splitting of $(10\bar{1})$ and $(101)$ reflections. Additionally, the absence of a small-angle $(010)$ reflection undoubtedly confirmed the true monoclinic symmetry and the real value of the $b$ parameter of the unit cell. In the course of our single-crystal structural study, it was supposed that the aparent orthorhombic unit cell of the sample was actually monoclinic, with the monoclinic angle being close to $90°$. Accordingly, the crystal structure was refined in an anisotropic approximation of thermal vibration in the space group $Pn$ as a two-component pseudomerohedric twin with a volume ratio of 32:68 and an obliquity of $0.01°$. Our structural data, obtained via a low-temperature single-crystal X-ray experiment, correlate with the findings of powder synchrotron studies, thereby providing undoubtable confirmation of the acentric space group $Pn$ inherent to the monoclinic modification of $Na_2CoSiO_4$.

Numerous studies have focused on the various physical properties of $Na_2CoSiO_4$ crystals, which exhibit good semiconductor properties and are promising for use in high-power Na-ion solar cells [20]. $Na_2CoSiO_4$ has also been investigated as a positive electrode material for sodium-ion capacitors. It showed excellent electrochemical performance, with a specific capacity of 42 F/g and a high specific energy of 12.4 Wh/kg (at a power density of 782.7 W/kg), and excellent cycling performance, retaining up to 84% of its capacity after 1500 charge–discharge cycles [18]. The results of [17] recommend the potential use of $Na_2CoSiO_4$ as a cathode material for sodium-ion batteries. Electrochemical measurements using cells with metallic sodium anodes revealed a reversible specific capacity of 100 mAh/g at an average discharge voltage of 3.3 V vs. $Na/Na^+$. The atomistic modeling of $Na^+$ diffusion demonstrated a low activation barrier and the presence of three-periodic diffusion pathways within the silicate framework, signifying favorable $Na^+$ intercalation kinetics.

### 3.4. $Na_2Cu_3O(Cu_{0.8}Na_{0.2})(PO_4)_2Cl$ with Oxo-Centered Pyroxene-like Chains

3.4.1. Analysis of Interatomic Distances and Description of the Crystal Structure

The key structural units of $Na_2Cu_3O(Cu_{0.8}Na_{0.2})(PO_4)_2Cl$ include an orthophosphate tetrahedron, two $Cu^{2+}$-centered polyhedra, and a $Cu^+$-centered trigonal bipyramid (Figure 4a). The $Cu1^{2+}O_6$ polyhedron with $2/m$ symmetry is a rare example of a compressed Jahn–Teller octahedron. There are four Cu1–O bonds with lengths equal to 2.254(2) Å and two shorter bonds of 1.851(1) Å (Table 5). The polyhedron distortion through the compressed axial oxygen atoms is associated with the requirement to avoid closer interaction between four neighboring $Cu2O_4Cl$ pyramids. The $Cu2^{2+}O_4Cl$ polyhedron is a distorted tetragonal pyramid, with the four Cu2–O distances at the base of the pyramid located in the range 1.911(2)–2.022(2) Å, and the Cl apex is located 2.579(1) Å from the Cu2.

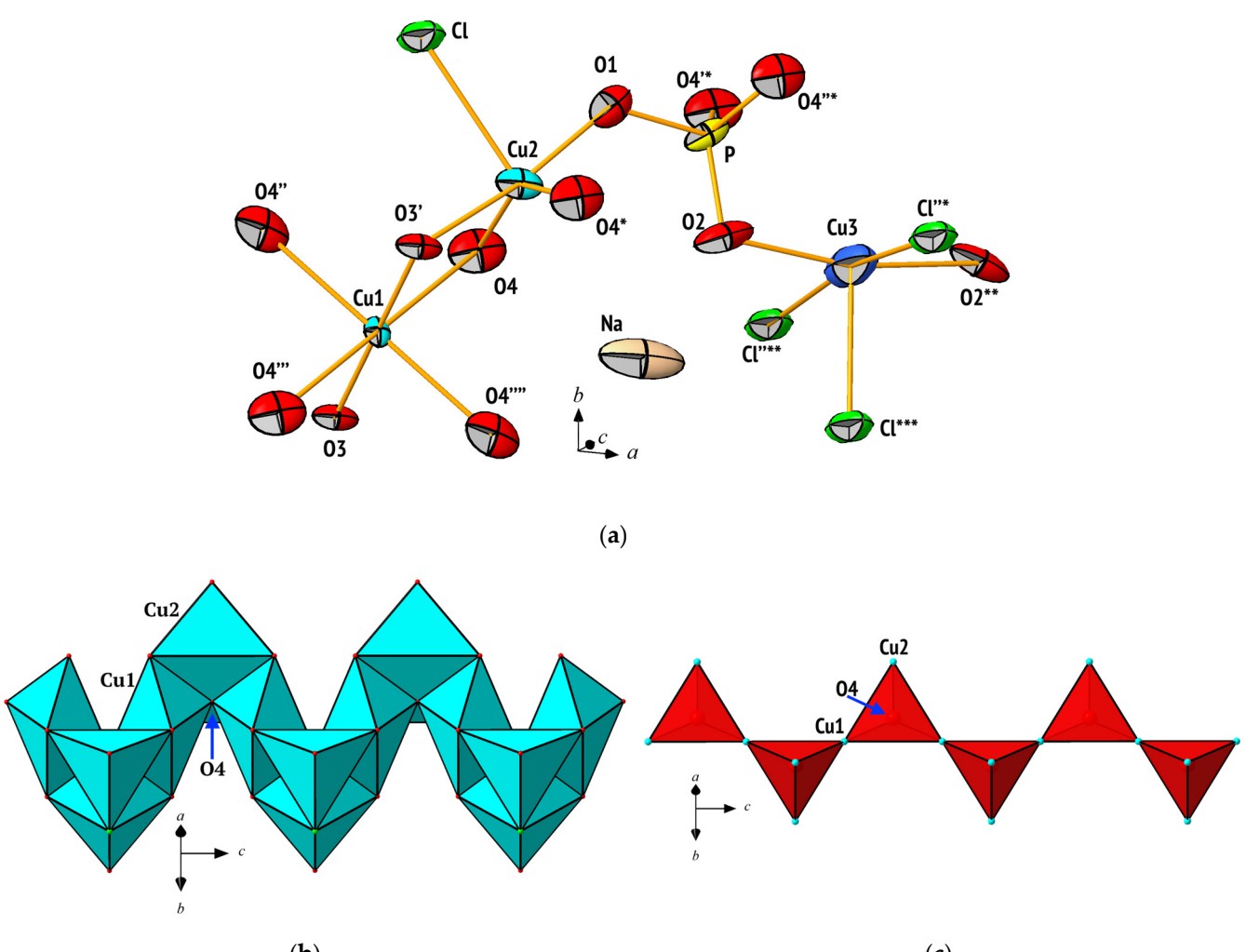

**Figure 4.** (**a**) The basic structural units of $Na_2Cu_3O(Cu_{0.8}Na_{0.2})(PO_4)_2Cl$ with an atom-labeling scheme. Displacement ellipsoids are represented at the 90% probability level. Symmetry codes: (′) 2 − *x*, 1 − *y*, and 0.5 + *z*; (″) 2 − *x*, *y*, and *z*; (‴) 2 − *x*, 1 − *y*, and 2 − *z*; (⁗) *x*, 1 − *y*, and 2 − *z*; (*) *x*, *y*, and 2.5 − *z*; (**) 3 − *x*, *y*, and 2.5 − *z*; (***) 0.5 + *x*, y − 0.5, and *z*; ('*) 2.5 − *x*, 1.5 − *y*, and 2 − *z*; (″*) 2.5 − *x*, 1.5 − *y*, and 0.5 + *z*; (‴**) 2.5 − *x*, 1.5 − *y*, and −0.5 + *z*. (**b**) Corrugated chain of $Cu^{2+}$-centered polyhedra in the classic representation and (**c**) the oxo-centered setting, showing $[O_2Cu_6]^\infty$ chains.

**Table 5.** The interatomic distances, Å, in the $Na_2Cu_3O(Cu_{0.8}Na_{0.2})(PO_4)_2Cl$ structure.

| Cu1 Octahedron | | Cu2 Tetragonal Pyramid | | Cu3 * Trigonal Bipyramid | |
|---|---|---|---|---|---|
| Cu1–O3 × 2 | 1.852(1) | Cu2–O3 | 1.911(2) | Cu3–O2 × 2 | 2.032(2) |
| –O4 × 4 | 2.254(2) | –O1 | 1.921(2) | –Cl | 2.394(5) |
| | | –O4 × 2 | 2.022(2) | –Cl × 2 | 3.1875(1) |
| <Cu1–O> | 2.12 | –Cl1 | 2.5789(8) | | |
| | | <Cu2–O> | 1.97 | <Cu3–O> | 2.03 |
| **Na Octahedron** | | | **P Tetrahedron** | | |
| Na–O2 × 2 | 2.309(2) | | P–O2 | | 1.532(2) |
| –O1 × 2 | 2.460(2) | | –O1 | | 1.544(2) |
| –O4 × 2 | 2.884(2) | | –O4 × 2 | | 1.544(2) |
| –Cl × 2 | 3.270(2) | | | | |
| <Na–O> | 2.55 | | <P–O> | | 1.54 |

* Cu3-site is occupied by 74% $Cu^+$ and 26% $Na^+$.

The $Cu^+O_2Cl_3$ polyhedron possesses *mm*2 symmetry and differs from the discussed $Cu^{2+}$-centered polyhedra. The Cu3 position is 74% occupied by $Cu^+$ cations and 26% occupied by $Na^+$ cations. The Cu3-site atoms are coordinated by the two nearest O atoms, with the distance between the Cu3 and O2 of 2.032(2) Å and $Cl^-$ anions, one at a distance of 2.394(5) Å and two at significantly larger distance of 3.1875(1) Å, resulting in trigonal bipyramidal coordination. The P–O bond lengths within the tetrahedron at the *m* plane range from 1.530(2) to 1.544(2) Å. The Na atom on the 2-fold axis is surrounded by eight ligands, i.e., six O anions with Na–O distances varying from 2.309 (1) to 2.886(2) Å and two Cl atoms at 3.270(2) Å (Figure 5a). This eight-fold coordination of the Na atoms is clearly confirmed via bond valence calculation (Table 6).

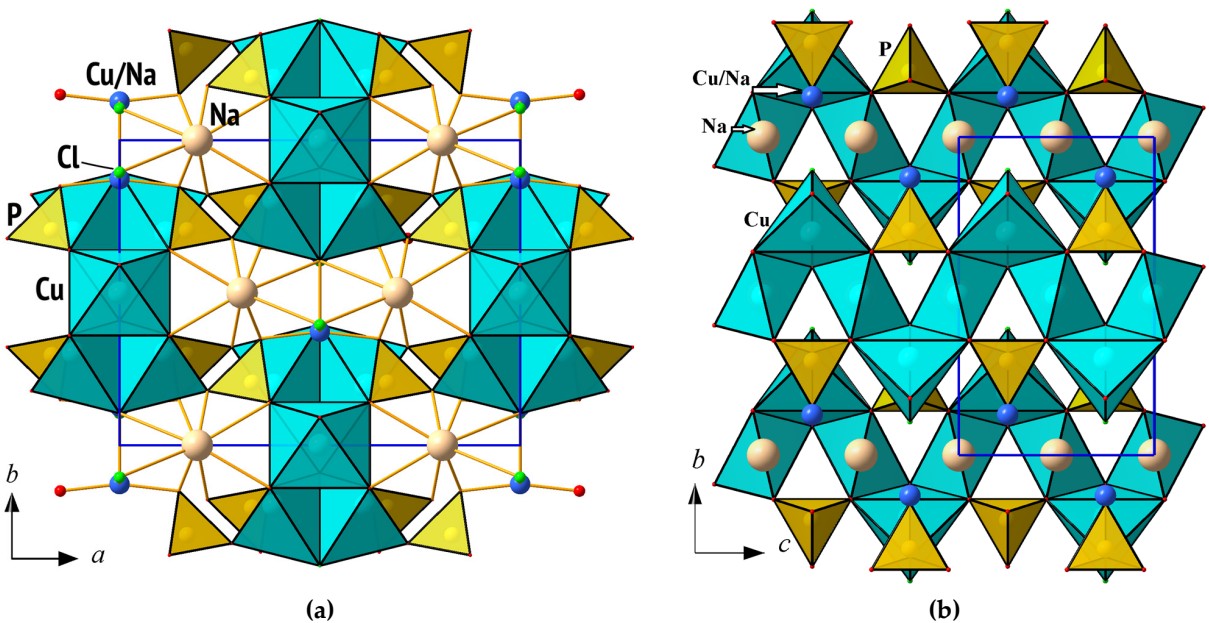

**Figure 5.** The crystal structure of $Na_2Cu_3O(Cu_{0.8}Na_{0.2})(PO_4)_2Cl$ displayed along the *z* (**a**) and *x* (**b**) axes.

**Table 6.** The bond valence calculations for $Na_2Cu_3O(Cu_{0.8}Na_{0.2})(PO_4)_2Cl$ *.

|  | **Cu1** | **Cu2** | **Cu3 \*\*** | **P** | **Na** | **∑** |
|---|---|---|---|---|---|---|
| O1 |  | 0.520 |  | 1.212 | 0.155 $^{×2↓ ×2→}$ | **2.04** |
| O2 |  |  | 0.182 $^{×2↓}$ | 1.251 | 0.212 $^{×2↓ ×2→}$ | **1.98** |
| O3 | 0.626 $^{×2↓ ×2→}$ | 0.534 $^{×2→}$ |  |  |  | **2.32** |
| O4 | 0.212 $^{×4↓}$ | 0.395 $^{×2↓}$ |  | 1.213 $^{×2↓}$ | 0.064 $^{×2↓}$ | **1.88** |
| Cl |  | 0.209 $^{×2→}$ | 0.144<br>0.02 $^{×2↓ ×2→}$ |  | 0.073 $^{×2↓ ×4→}$ | **1.04** |
| **∑** | **2.10** | **2.05** | **0.95** | **4.89** | **1.01** |  |

\* Symbols $^{×2↓}$ and $^{×2→}$ indicate the multiplication of the valence contribution along the column or row according to the symmetry. \*\* Calculated as occupied by 74% $Cu^+$ and 26% $Na^+$.

In the crystal structure of the compound $Na_2Cu_3O(Cu_{0.8}Na_{0.2})(PO_4)_2Cl$, $Cu^{2+}$–centered polyhedra form ribbons, in which each $CuO_6$ octahedron shares four edges with four $CuO_4Cl$ pyramids (Figure 4b). Likewise, each pyramid shares three edges, one edge with the neighboring $CuO_4Cl$ polyhedron and two edges with the $CuO_6$ octahedra. The Cl vertex is common to two $Cu2O_4Cl$ polyhedra, and the oxygen ligand O4 is common to four Cu polyhedra (two $CuO_4Cl$ and two $CuO_6$). Thus, two types of $Cu^{2+}$-centered polyhedra form condensed zigzag ribbons elongated in the direction of the *c*-axis of the unit cell. These

ribbons are arranged along the midpoints of the lateral faces of the unit cell in accordance with the base-centered lattice (oS). Orthophosphate tetrahedra connect these ribbons into a three-periodic heteropolyhedral framework (Figure 5). Similar ribbons are described in the crystal structures of the mineral aleutite ($(M_{0.5}Cl)[Cu_5O_2(AsO_4)(VO_4)]$) [21] and the synthetic phase ($(Na,Li)_3(Cl,OH)[Cu_3OAl(PO_4)_3]$) [22].

Each $PO_4$ group shares three oxygen vertices with $Cu^{2+}$-centered polyhedra; hence, the $PO_4$ tetrahedron connects two neighboring copper ribbons along the *a*-axis. The fourth oxygen vertex of the $PO_4$ group is directed inside the framework channels and involved in the coordination of $Na^+$ and $Cu^+$ ions. The presence of these monovalent cations inside the large channels compensates for the negative charge of the heteropolyhedral framework.

### 3.4.2. $Na_2Cu_3O(Cu_{0.8}Na_{0.2})(PO_4)_2Cl$ in the Series of Compounds with Oxo-Centered Pyroxene-like Chains

As mentioned above, the $Na_2Cu_3O(Cu_{0.8}Na_{0.2})(PO_4)_2Cl$ crystal structure is based on the chains of interconnected $CuO_6$ octahedra and $CuO_4Cl$ pyramids, sharing edges. In these chains, neighboring Cu atoms are located at distances of 3.0264(3) Å (Cu1–Cu2) and 3.1382(6) Å (Cu2–Cu2) from each other. The oxygen ligand O4 in each of these chains is exclusively surrounded by copper atoms, forming the $\mu_4$ configuration.

According to the established nomenclature, copper oxo-salts with "additional" oxygen ligands (not shared with anionic complexes) are called oxo-cuprates. They are often described within the anion-centered approach [23,24]. In the case of the $Na_2Cu_3O(Cu_{0.8}Na_{0.2})$ $(PO_4)_2Cl$, oxo-centered $[OCu_4]$ tetrahedra form pyroxene-like chains $[O_2Cu_6]_\infty$ (Figure 4c) that belong to the "zweier chain topological type", according to the classification in [24]. In the structure **(II)**, these chains, extended along the *c*-axis of the unit cell, are separated from each other by Na-, P-, and $Cu^+$-centered polyhedra.

It is noteworthy that the $Na_2Cu_3O(Cu_{0.8}Na_{0.2})(PO_4)_2Cl$ is isostructural to the oxo-cuprate $Na_2Cu_3OCu(PO_4)_2Cl$, in the crystal structure of which there is no $Cu^+/Na^+$ iso-morphous substitution [25]. Additionally, in the same family of electronically distinct copper phosphate chlorides, an oxo-cuprate arsenate $Na_2Cu_3OCu(AsO_4)_2Cl$ was recently obtained [26]. This phase crystallizes in the lower symmetry space group *Pnma*, which is a subgroup of the *Cmcm* group inherent to phosphate.

Other structurally related phases, i.e., $NaCu_3OCu(PO_4)_2Cl$ [27] and $Cu_{1.5}Cu_3O(PO_4)_2$ $Cl$ [26], only contain divalent copper cations. All discussed structures are formed from the same ribbon fragments of $Cu^{2+}$-centered polyhedra, interconnected through anionic tetrahedra. The differences between the structures lie in variations in the channel content within the topologically identical crystal frameworks of the composition $[(Cu_3O)(XO_4)_2]^{2-}{}_\infty$, where *X* denotes P or As. In the $Na_2Cu_3O(Cu_{0.8}Na_{0.2})(PO_4)_2Cl$ crystal structure, the channels contain salt inclusions of the $|Na_2(Cu^I_{0.8}Na_{0.2})Cl|^{2+}$ composition. In the arsenate structure, there is a similar inclusion of $|Na_2Cu^ICl|^{2+}$, whereas phases containing only divalent copper cations comprise $|NaCu^{II}Cl|^{2+}$ and $|Cu^{II}_{1.5}Cl|^{2+}$. The channels contents differ in the amount of sodium cations, as well as the atomic positions of the interstitial ions. Notably, similar salt inclusions, often found in host–guest structures, are also observed in oxo-cuprates synthesized through halide flux methods, such as $Na_2Cu_3OCu(PO_4)_2Cl$ [25] and $NaCu_3OCu(PO_4)_2Cl$ [27]. However, hydrothermal synthesis techniques are also suitable for the preparation of these compounds, as shown in [22,28].

It is remarkable that anion-centered oxo-cuprate chains ($[O_2Cu_6]_\infty$) (Figure 4c) form not only the crystal structure of **(II)** but also the other synthetic compounds mentioned above. These oxo-centered fragments ($[O_2Cu_6]_\infty$) are characteristic of minerals found in fumaroles, namely kamchatkite ($KCu_3OCl(SO_4)_2$) [29,30], chloromenite ($Cu_9O_2(SeO_3)_4Cl_6$) [31], vergasovaite ($Cu_3O[(Mo,S)O_4SO_4]$) [32], dokuchaevite ($Cu_2[Cu_6O_2](VO_4)_3Cl_3$) [33], yaroshevskite ($Cu_3[Cu_6O_2][VO_4]_4Cl_2$) [34], and cupromolybdite ($[O_2Cu_6][MoO_4]_4$) [35]. Nevertheless, in the structures of minerals, tetrahedra forming oxo-centered $[O_2Cu_6]_\infty$ chains have an orientation different to those in the structures of synthetic phases [26].

### 3.5. Mineralogically Probable Na₃V(OH)(HPO₄)(PO₄)

3.5.1. Description of Crystal Structure and Analysis of Interatomic Distances

In the crystal structure of the $Na_3V(OH)(HPO_4)(PO_4)$ compound, V-centered octahedra share O2 vertices with the hydroxyl group to form chains of the $2_1$ symmetry, parallel to the *b* axis of the unit cell. Each $PO_4$ tetrahedron shares two O vertices adjacent to the chain $VO_4(OH)_2$ octahedra, resulting in the arrangement of one-periodic mixed-type anionic fragments of $[V(OH)(HPO_4)(PO_4)]^{3-}$. These negatively charged ribbons interconnect in a framework via $Na^+$ ions and hydrogen bonds (Figure 6). The hydrogen bonds system includes the atom O1, which serves as a donor to form an O1—H1·····O6 bond of medium strength, in accordance with the $D\cdots A$ distance (Table 7). The O1 atoms also acts as an acceptor in the significantly weaker hydrogen bond O2—H2·····O1 (Figure 6b).

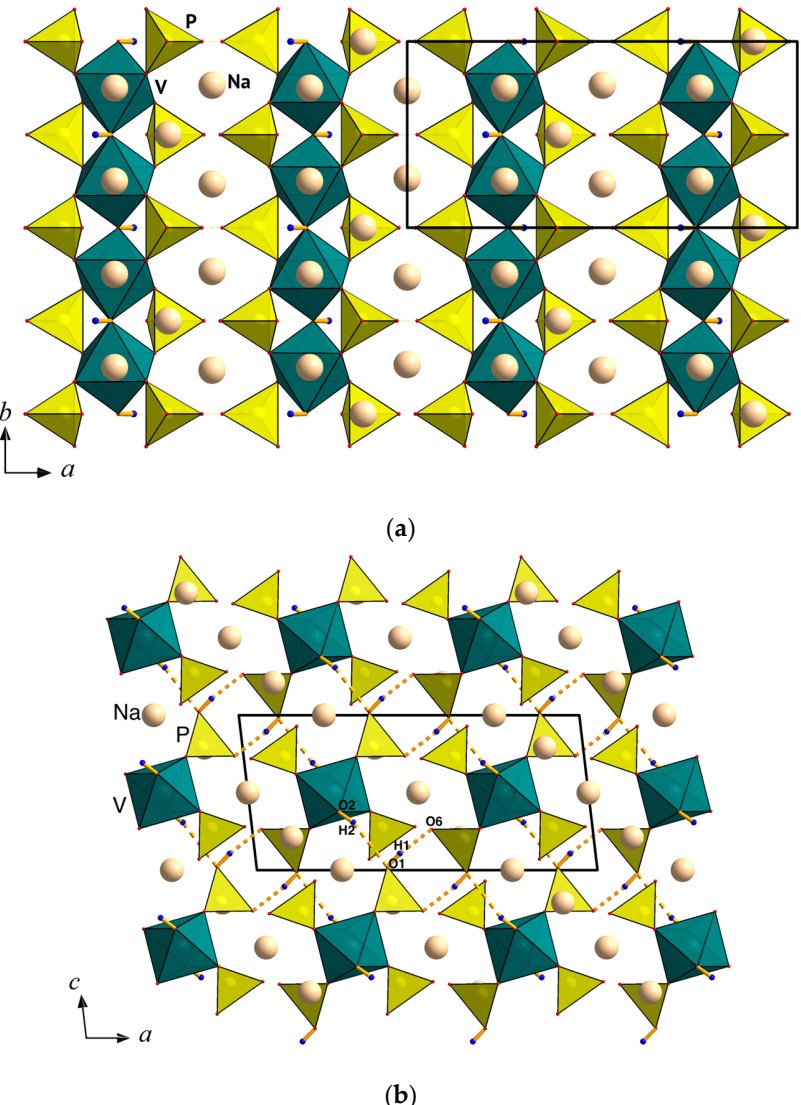

(a)

(b)

**Figure 6.** Ribbons of vanadium octahedra surrounded by phosphate tetrahedra (**a**) and their hydrogen bonding (**b**) in the $Na_3V(OH)(HPO_4)(PO_4)$ crystal structure projected onto the *ab* and *ac* planes.

**Table 7.** The geometric characteristics of hydrogen bonds in the $Na_3V(OH)(HPO_4)(PO_4)$ structure.

| *D*–H⋯*A* | *D*–H, Å | H⋯*A*, Å | *D*⋯*A*, Å | *D*–H⋯*A*, ° |
|---|---|---|---|---|
| O1–H1⋯O6 | 0.82(1) | 1.80(2) | 2.616(7) | 172(9) |
| O2–H2⋯O1 | 0.82(1) | 2.53(3) | 3.319(7) | 164(8) |

The basic structural units of the Na$_3$V(OH)(HPO$_4$)(PO$_4$) crystal structure are two symmetry-independent orthophosphate tetrahedra and a V$^{3+}$O$_4$(OH)$_2$ octahedron, as shown in Figure 7. Both the phosphate tetrahedra with *m* symmetry exhibit significant distortion. The bond lengths in the P2O$_4$ tetrahedron vary from 1.508(5) to 1.558(3) Å (Table 8). The shortened P2–O5 and P2–O7 bond lengths equal to 1.508(5) Å and 1.510(5) Å, respectively, serve to balance the valence charge on the corresponding oxygen atoms (Table 9). The P1O$_3$(OH) tetrahedron is characterized by a notably longer P1–O1 distance of 1.582(5) Å to the oxygen atom of the hydroxyl group compared to other P1–O bond lengths ranging from 1.526(5) Å to 1.538(3) Å. Both the average P–O distances associated with P1 and P2 tetrahedra are equal to 1.534 and 1.544 Å, which are usual for the orthophosphate groups (Table 8). The V$^{3+}$ ion centers the centrosymmetric V$^{3+}$O$_4$(OH)$_2$ octahedron characterized by V–O bond distances within the range 1.966(3)–2.039(2) Å, with an average value of 2.01 Å, which is typical for V$^{3+}$ cations. The structure also contains three symmetrically independent sodium atoms, forming centrosymmetric octahedra Na1O$_4$(OH)$_2$ and Na3O$_6$, and a trigonal bipyramid Na2O$_5$. In these polyhedral, the Na–O distances vary from 2.175(6) to 2.550(3) Å, while the Na1–OH values are expectedly larger and equal to 2.747(4) Å.

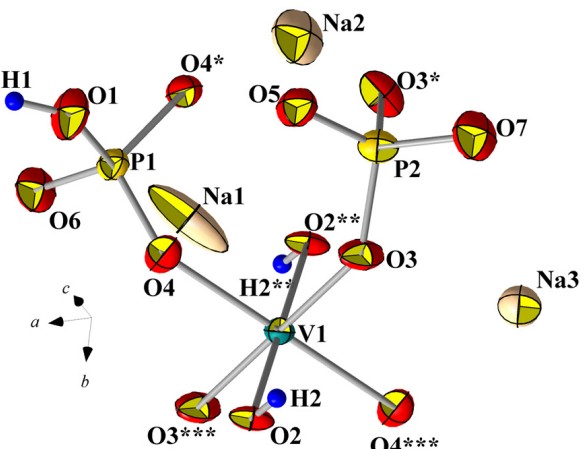

**Figure 7.** An independent fragment of the crystal structure of Na$_3$V(OH)(HPO$_4$)(PO$_4$) **(III)** with thermal ellipsoids represented at the 90% probability level. Symmetry operations: (*) $x$, $1 - y$, $z$; (**) $1.5 - x$, $-0.5+y$, $-z$; (***) $1.5 - x$, $1.5 - y$, $-z$.

**Table 8.** The interatomic distances (Å) in the crystal structure of Na$_3$V(OH)(HPO$_4$)(PO$_4$).

| V-Centered Octahedron | | P1–Tetrahedron | | P2–Tetrahedron | |
|---|---|---|---|---|---|
| V–O3 | 1.966(3) | | | | |
| –O3 | 1.966(3) | P1–O6 | 1.526(5) | P2–O5 | 1.508(5) |
| –O4 | 2.020(3) | –O4 | 1.538(3) | –O7 | 1.510(5) |
| –O4 | 2.020(3) | –O4 | 1.538(3) | –O3 | 1.558(3) |
| –O2 | 2.039(2) | –O1 | 1.582(5) | –O3 | 1.558(3) |
| –O2 | 2.039(2) | <P1–O> | 1.544 | <P2–O> | 1.534 |
| <V–O> | 2.008 | | | | |
| **Na1–Octahedron** | | **Na2–Polyhedron** | | **Na3–Octahedron** | |
| Na1–O5 | 2.259(3) | | | Na2–O3 | 2.427(3) |
| –O5 | 2.259(3) | Na2–O7 | 2.175(6) | –O3 | 2.427(3) |
| –O4 | 2.448(3) | –O5 | 2.238(6) | –O6 | 2.460(4) |
| –O4 | 2.448(3) | –O2 | 2.435(5) | –O6 | 2.460(4) |
| –O1 | 2.747(4) | –O4 | 2.550(3) | –O7 | 2.482(4) |
| –O1 | 2.747(4) | –O4 | 2.550(3) | –O7 | 2.482(4) |
| <Na1–O> | 2.485 | <Na2–O> | 2.390 | <Na3–O> | 2.456 |

**Table 9.** The bond valence data for $Na_3V(OH)(HPO_4)(PO_4)$.

| | Na1 | Na2 | Na3 | V | P1 | P2 | H1 | H2 | Σ |
|---|---|---|---|---|---|---|---|---|---|
| O1 | 0.086 $^{×2↓ ×2→}$ | | | | 1.093 | | 0.71 | 0.06 | **2.04** |
| O2 | | 0.163 | | 0.449 $^{×2↓ ×2→}$ | | | | 0.94 | **2.00** |
| O3 | | | 0.166 $^{×2↓}$ | 0.547 $^{×2↓}$ | | 1.167 $^{×2↓}$ | | | **1.88** |
| O4 | 0.159 $^{×2↓}$ | 0.129 $^{×2↓}$ | | 0.473 $^{×2↓}$ | 1.231 $^{×2↓}$ | | | | **1.99** |
| O5 | 0.235 $^{×2↓ ×2→}$ | 0.246 | | | | 1.335 | | | **2.05** |
| O6 | | | 0.155 $^{×2↓ ×2→}$ | | 1.272 | | 0.29 | | **1.87** |
| O7 | | 0.28 | 0.148 $^{×2↓ ×2→}$ | | | 1.328 | | | **1.90** |
| Σ | **0.96** | **0.95** | **0.94** | **2.94** | **4.83** | **5.00** | **1** | **1** | |

The symbols $^{×2↓}$ and $^{×2→}$ indicate the multiplication of the valence contribution along a column or row in accordance with symmetry.

The crystal structure of the compound with the chemical formula $Na_3V(OH)(HPO_4)(PO_4)$ was first published in 2008 in [36], the authors of which used X-ray powder data in order to obtain the structure model. In our study based on a single-crystal X-ray diffraction experiment, the model was confirmed, and hydrogen atoms were also located and their positions were refined. Moreover, our structural study provides precise data regarding the geometry of $PO_4$ tetrahedra, which allows us to exclude disorder in P-sites, as supposed in [36].

It is notable that the compound $Na_3V(OH)(HPO_4)(PO_4)$ has isostructural analogues with $V^{3+}$ substituted for $Al^{3+}$ or $Ga^{3+}$ [37], which, together, form a group of compounds with the general formula $Na_3(M)(OH)(HPO_4)(PO_4)$, where $M$ = V, Al, or Ga.

Similar crystal chemical properties of $Fe^{3+}$ and $V^{3+}$ ions determine a widespread distribution of vanadium in the magmatic process. Thus, iron is a kind of solvent of trivalent vanadium, causing its dispersed state in igneous rocks. In hydrothermal solutions in a reducing environment, vanadium migrates in the form of $(V^{4+}O)^{2+}$ or $V^{3+}$ ions, which are mainly precipitated as part of hydrated silicate minerals, such as dimorphic varieties of cavansite and pentagonite, i.e., $Ca(V^{4+}O)Si_4O_{10}·4H_2O$. Though the $(V^{4+}O)^{2+}$ or $V^{3+}$ phosphates are rare, among them are sincosite $(Ca(V^{4+}O)_2(PO_4)_2·5H_2O)$, and bariosincosite, $(Ba(V^{4+}O)_2(PO_4)_2·4H_2O)$, and a unique $V^{3+}$ representative known as springcreekite $(BaV^{3+}_3(PO_4)(PO_3OH)(OH)_6)$. Both springcreekite and bariosincosite, found in a small vein deposit formerly mined for Cu, are considered to be formed via subaerial and near surface aqueous alteration at low temperatures [38,39]. It is remarkable that sincosite and bariosincosite synthetic analogues have been obtained in the laboratory under similar P/T conditions [40–42]. The geochemically reasonable system for $Na_3V^{3+}(OH)(HPO_4)(PO_4)$ synthesis at 270 °C and 70 atm, as well as the discovery of natural $V^{3+}$ phosphate springcreekite, indicate the mineralogical probability of this compound.

3.5.2. Magnetic Properties of $Na_3V(OH)(HPO_4)(PO_4)$

$V^{3+}$ has a $3d^2$ electronic configuration. The low value of spin $S$ = 1 in combination with the reduced dimensionality of the magnetic subsystem can be a source of quantum behavior at low temperatures. Previously, the magnetic behavior of $Na_3V(OH)(HPO_4)(PO_4)$ was studied in the temperature range of 50–250 K. Magnetic susceptibility measurements evidenced sizable exchange interactions in the title compound. Therefore, it was interesting to study its behavior down to lower temperatures.

Magnetization $M$ was measured using a Quantum Design automated PPMS-9T (Physical Property Measurement System) equipped with a Vibrating Sample Magnetometer (VSM). The temperature dependences of the magnetic susceptibility were taken in field-cooled (FC) and zero-field-cooled (ZFC) regimes at $B$ = 0.1 T in the temperature range of 2–300 K. Through the temperature region, ZFC and FC curves almost perfectly coincided

(Figure 8). At elevated temperatures, the $\chi(T)$ curves followed the Curie–Weiss law with the addition of the following temperature independent term $\chi_0$:

$$\chi = \chi_0 + \frac{C}{T - \Theta} \tag{1}$$

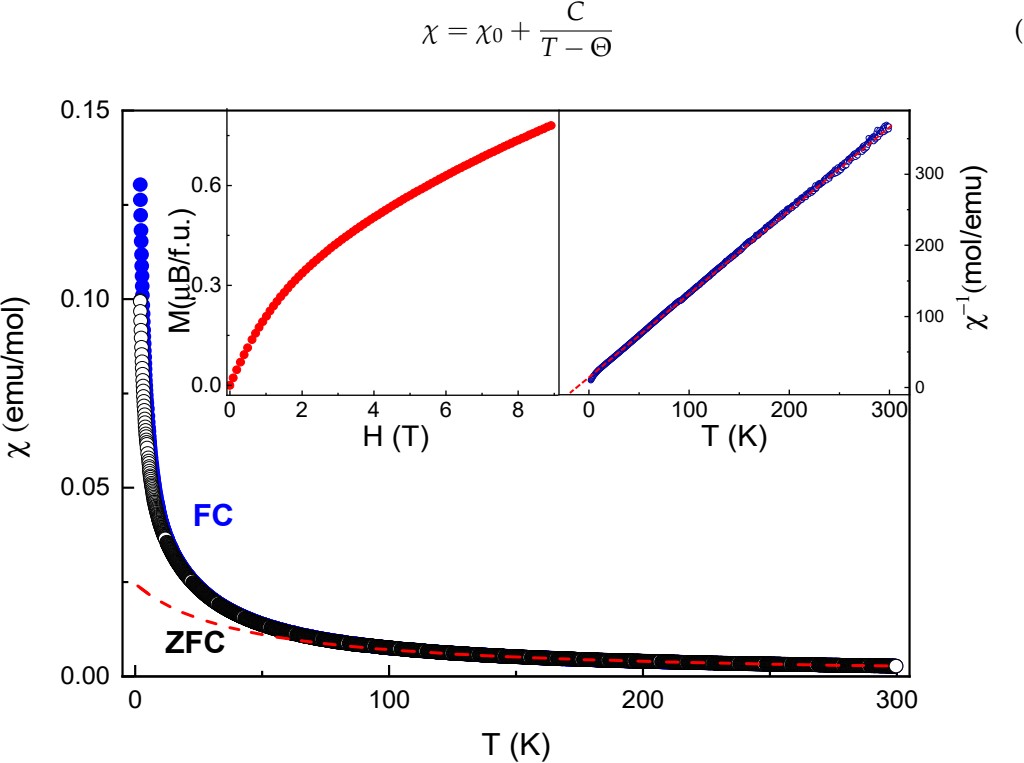

**Figure 8.** The temperature dependences of the magnetic susceptibility in $Na_3V(OH)(HPO_4)(PO_4)$ at $B = 0.1$ T taken in field-cooled (blue circles) and zero-field-cooled (white outlined circles) regimes. The dash red line illustrates the extrapolation of the Curie–Weiss law. The left inset presents the magnetization curve at $T = 2$ K, while the right inset shows the inverse magnetic susceptibility and its extrapolation to a negative temperature (dash red line).

The fitting of the FC curve in the range 200–300 K results in the Curie constant $C = 0.99$ emu K/mol and Weiss temperature $\Theta = -35$ K. The value of $C$ corresponds to the presence of $V^{3+}$ ions in the structure using the $g$-factor $g = 1.99$. The negative Weiss temperature $\Theta$ suggests antiferromagnetic correlations between $V^{3+}$ cations. The right inset in Figure 8 presents inverse magnetic susceptibility, showing deviation from linearity near 10 K. The extrapolation of $\chi^{-1}(T)$ curve to negative temperatures indicates a slightly lower value of Weiss temperature, established via the approximation of the high-temperature region of the $\chi(T)$ curve.

The magnetization curve (left inset in Figure 8) reveals that at $T = 2$ K and in a magnetic field of $B = 9$ T, the magnetization is 0.78 $\mu_B$. This value is significantly lower than that of the saturation magnetization, calculated using the following formula: $M_{sat} = ngS\mu_B$, which is equal to 2 $\mu_B$.

According to the slightly elongated geometry of $VO_4(OH)_2$ polyhedra, two $d$ electrons of $V^{3+}$ occupy two out of three $t_{2g}$ orbitals. The established presence of antiferromagnetic exchange interactions between V-centered octahedra within chains is supported by the bridge angle V–O–V = 127.5° obtained from structural data. The absence of evidence for low-dimensional behavior in magnetic susceptibility measurements allows us to suggest that the values of exchange magnetic parameters of intrachain and interchain interactions are comparable in $Na_3V(OH)(HPO_4)(PO_4)$. Lower temperatures are needed to achieve a long-range order in the title compound.

## 4. Conclusions

In our search for the compounds with alkali and transition metals, three mineral-like phases were synthesized in the form of single crystals: silicate $Na_2CoSiO_4$, a structural analogue of liberite, and two phosphates, namely $Na_2Cu_3O(Cu,Na)(PO_4)_2Cl$ and $Na_3V(OH)(HPO_4)(PO_4)$. Their crystal structures were refined (partially based on low temperatures) via X-ray diffraction, unlike some earlier studies performed using the powder samples. Our research, which generally confirmed previous results, allowed us to obtain more precise atomic coordinates, distances, and hydrogen bonds. It was shown that the silicate $Na_2CoSiO_4$ is characterized by a strongly pseudo-orthorhombic lattice and a monoclinic crystal structure, refined using a pseudomerohedric microtwin. It was established that in the crystal structure of $Na_2Cu_3O(Cu,Na)(PO_4)_2Cl$, the $Cu^+$ position is partially substituted by 26% Na atoms, in contrast to previous structural data. We also highlighted the original crystal–chemical interpretation of the structural features and properties of $Na_2CoSiO_4$ and $Na_2Cu_3O(Cu,Na)(PO_4)_2Cl$ and the studied magnetic behavior of $Na_3V(OH)(HPO_4)(PO_4)$.

**Supplementary Materials:** The following supporting information can be downloaded at: https://www.mdpi.com/article/10.3390/min14010046/s1, cif and checkcif files.

**Author Contributions:** Conceptualization, O.Y. and G.K.; methodology, O.D. and A.V.; software, O.Y. and G.K.; investigation, G.K., O.Y., P.V., A.V., O.D., S.S. and L.S.; writing—original draft preparation, G.K., P.V. and L.S.; writing—review and editing, O.Y.; visualization, G.K., P.V., O.Y. and L.S.; supervision, O.Y. All authors have read and agreed to the published version of the manuscript.

**Funding:** This work was supported by Moscow Lomonosov State University, Russian Federation (award no. AAAA-A16-116033010121-7). A. V. thanks the Russian Science Foundation, grant no. 23-73-10125, for providing financial support. The study by S.S. was carried out as part of the state assignment for the ISSP RAS.

**Data Availability Statement:** We deposited structural data via the joint CCDC/FIZ Karlsruhe deposition service under the deposition numbers 2310942 **(I)**, 2310943 **(II)**, and 2310944 **(III)**. Cif-data, accessed on 29 November 2023, can be obtained free of charge from FIZ Karlsruhe via the following webpage: www.ccdc.cam.ac.uk/structures, accessed on 29 November 2023.

**Acknowledgments:** We thank N. N. Koshlyakova for carrying out the microprobe analysis of the crystals. We are grateful to N. V. Zubkova for providing assistance in collecting diffraction data, and we thank A. N. Vasiliev for consultations on magnetism.

**Conflicts of Interest:** The authors declare no conflict of interest. The funders had no role in the design of the study; the collection, analyses, or interpretation of data; the writing of the manuscript; or the decision to publish the results.

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
