# Peer review of "Mineral-like Synthetic Compounds Stabilized under Hydrothermal Conditions: X-ray Diffraction Study and Comparative Crystal Chemistry"

_minerals, doi:10.3390/min14010046_

Round 1

Reviewer 1 Report

Comments and Suggestions for Authors

This is an excellent paper on the synthesis, crystal structure, and magnetic properties of three transition-metal oxysalts, one of which could potentially be found in nature and the other two of which have already been discovered as minerals. It presents not only their synthesis route and structure description, as well as some magnetic properties, but also an overview and comparison of these structures with structurally similar compounds.

The only minor change I would recommend is rewriting the introduction, which is far too broad and long for such an article.

Reviewer 2 Report

Comments and Suggestions for Authors

The motivation for doing the work is clearly stated and well justified. My main criticism is that the Introduction should be shortened and more concisely expressed.

There is also undue repetition between the Conclusions and the Introduction. The Conclusions should likewise be shortened and referred back to the Introduction.

Otherwise the quality of the graphics is high and the structural discussion well founded. As an English native speaker I am qualified to recommend the following improvements to the Scientific English. Further improvements can be carried out by a Copy Editor.

L. 74 Jahn-Teller

L. 81 mineralizers

L. 98 sodium hydroxide

L. 106 „by light microscopy“ instead of „under binoculars“

L. 116 with Na:V:P ratios equal to 3:1:2

L. 117 images

L. 128 „to“ instead of „up to“

L. 131 „whilst“ or „whereas“ instead of „while“

L. 140 The formulae refined…

L. 141 „We deposited…“ bad style. Use passive: „...were deposited“

L. 180-181: problematical sentence:

These three-periodic cristobalite-like constructions of sharing vertices NaO4 180 tetrahedra, on one hand, and CoO4/PO4 tetrahedra, on the other hand, interconnect to 181 each other, while look quasi-layered in the (101) projection“

The following is suggested:

The 3D cristobalite-like vertex-sharing of NaO4 tetrahedra and interconnected CoO4/PO4 tetrahedra leads to quasi-layering in (101) projection.“

L: 197: „ordered, unlike in the orthorhombic polymorph“ instead of „ordered opposite to the orthorhombic one“

L. 248 „requirement“ instead of „commitment“

L. 253 „Cu3 atom“

L. 380 „we“: bad style: use passive form of verb

L. 388: 3d2 electronic configuration

L. 393 „down to lower temperatures“

Comments on the Quality of English Language

See above comments in main section.

Reviewer 3 Report

Comments and Suggestions for Authors

This is pleasure for me to review interesting material described in this paper.

However, I think that authors choice on the text organization is wrong since it is almost impossible to distinguish the results obtained in this work from some other obtained previously. This is because the structure of the article does not correspond to standard article structure: Introduction - Materials and Methods - Results - Discussion - Conclusion. I was surprised to see some results in the section Materials and Methods. So, it is very hard to evaluate the material in this form. I strongly recommend text reorganization. Apart from that the text is not fully consistent and coherent. Some phrases sound like a jargonism. Although I note that minor English corrections are needed, moderate scientific(!) English correction are needed.  Taking into account authors`s experience I think that these corrections can be made by themself without additional services. In general, one gets the impression that the authors were in a hurry to submission, as a result of which the presentation suffered.

I do hope that authors will do rigorous work on the text, because such presentation of there material devalues their experimental work and makes it very difficult for the reader uninitiated in all the structural nuances to perceive.

I was also unpleasantly surprised by the absence of some citations in the text. For example, mentioning the structures of minerals without citing these works, while the authors did not forget to cite their work on synthetic materials. Work with literature also needs to be carried out more carefully.

Some examples are given below, but the whole text need work with reorganization and paraphrasing.

Introduction is not well organized from Earth Science perspective because hydrothermal and fumarolic processes are mixed up. For example authors state "The probability of discovering these crystal phases in natural geological settings depends on the geochemically sound systems used in laboratory experiments, as well as adequate P/T parameters and mineralizes emulating the natural environments. In this paper, we present experimental results of crystallization in hydrothermal systems that mimic expected middle-temperature hydrothermal conditions". However above information on fumarolic minerals (gas-transport reaction) is mentioned. The text is not coherent in that sense. The Introduction should be better organized in general. If it is about chemistry, ok, than do not mention mineral-inspired and mineral-like processes like potential discovering of these phases. Moreover nothing is mentioned about middle-temperature hydrothermal processes. Where does it occur? Examples? it does not have any (geological connection) to fumaroles...

Line 41 references should be added.

Line 105 somewhere here authors need to state that the resultant materials are listed in table 1. And that the phase notation is as I - , II - , III.

Lines 112-116 these are results. List standards used here.

Line 128 R1

Lines 130-131, which coordination is meant?

Lines 133-134 R-factor can hardly characterize anything. Delete. You have this value in the table.

Line 137 two symmetrically independent OH groups

Line 139 R1

Line 140-141 At that stage this is not convincing for me. I have not yet seen results of microprobe analysis. Standard phrase with not much sense.

Line 143 I think that commas in the cif numbers are misleading despite journal requirements (and should be omitted for this case).

Line 148 in this section Result? State it clearly.

Line 150 I would appreciate if authors start with some general information as structure type and structure motif. Which polyhedra are presented and how are they connect. With more details on Wyckoff positions etc. after that. Imagine that you have no idea on what is the obtained structure and you read this explanation...

Lines 165-166 Which additional O atoms are meant here? and why they are not included to the coordination sphere of Na? If I understand it correctly, there are O1-4 in the structure and all of them are already involved in Na-O bonding. Interestingly O1-4 have v.u. as nearly 2. What could be the source of additional v.u.??? May the site be partially vacant? What about chemical composition?

Line 181-182 "while look" and phrases like that should be avoided. What is meant under the term quasi-layer?

Line 204 Any charge, any size? U? K? Specify.

Lines 224-226 Here you mention comparison of your and previously published data. Much more information is needed. What is the confirmation of that space group? The reciprocal space? Do you also observe the splitting 221 of (101) and (101) reflections? Where we can see that? Or a small-angle (010) reflection? Prove this correlation by clear experimental findings.

Lines 253-254 Rewrite like The Cu3 site is occupied by .... The Cu3 site is coordinated by .... XX at the distance and XX at the distance resulting in trigonal .... Phrases like diluted etc should be avoided.

Line 257 where I can see Na complete coordination? In fig 4 coordination is incomplete. rewrite like Na tom is located and coordinated by 8 ligands: six O atoms at the distance .... and 2 Cl atoms at the distance ...
The participation of Cl ions in Na coordination are evident by bond valences (Table 6) with O1-4 contribution as XX v.u. and Cl contribution as XX v.u.

Line 273 why this is the main structural fragment?

Line 283 reference for aleutite is mandatory here. Siidra, O.I., Nazarchuk, E.V., Agakhanov, A.A., Yury S. Polekhovsky, Y.S. (2019) Aleutite [Cu5O2](AsO4)(VO4)·(Cu0.5â–¡0.5)Cl, a new complex salt-inclusion mineral with Cu2+ substructure derived from Kagome-net. Mineralogical Magazine: 83(6): 847-853.

Line 301 reference(s) for anion-centered approach.

Line 317 avoid our

Line 324 references for Na2Cu3OCu(PO4)2Cl and NaCu3OCu(PO4)2Cl

Lines 328-331 references for kamchatkite, chloromenite, vergasovaite, dokuchaevite, and yaroshevskite.

Line 338 Again, it would be better to start with broader characteristic of the structure. By mentioning basic units, then structure motif and only then PO4 geometries.

Lines 338-339 You mention both tetrahedra, but then give information only about one of them. Which tetrahedra are meant here? paraphrase and divide into two clear sentences.

Line 342-343 P1 atom id 4-coordinated with three bonds being in the range and one notably longer P1–O1 bond that is the evidence that O1 is the donor of O-H bonding(?)

Lines 344 and 345 what is meant under "Both average P–O distances"? Write it clear like both average P–O distances associated to P1 and P2 sites ...

Line 350 "delete with 1 symmetry"

Line 351 normally or in your structure?

Line 357 (A) or in A

Table 7 V-centered octahedra or V3+O4(OH)2 octahedron

Line 367 how do we know about "medium strengths"? Add "in accord with bond D-H...A lengths (Table 9)".

Line 377 paraphrase The crystal structure of compound with the chemical formula Na3V(OH)(HPO4)(PO4) was first published in 2008 by 37, the authors used X-ray powder diffraction data in order to obtain the structure model.

In our study we confirm the model by localization of XXXX atoms. In addition to that the study reveal the position of H atoms for the first time. (Do not use naturally, you need to explain everything; I suggest avoiding this sentence).

Line 334 I do not see the confirmation of the section Title in the text. The section is named "Mineralogically probable Na3V(OH)(HPO4)(PO4)", but nothing said in the text about mineralogical probability.

Lines 384-386 what about localization of H atoms in these compounds?

Noteworthy, that the compound Na3V(OH)(HPO4)(PO4) has isostructural analogs with V3+ substituted by Al3+ or Ga3+, hence the group of compounds with the general formula Na3(M)(OH)(HPO4)(PO4), where M = Al or Ga [38].

Lines 417-418 ? (citation) allows suggesting that intrachain interactions in Na3V(OH)(HPO4)(PO4) are comparable to interchain interactions.

Conclusions lines 427-447 only the new findings should remain in this section. Delete lines 427-437. Make conclusions more precise (very brief results and their interpretation that you would like to stress, they are too broad now.

It seems like response to Alert B is incomplete "Author Response: Thermal ellipsoids of the Cu are elongated because Cu(I)atoms occupy t".

I hope this would increase the manuscript quality.

Comments on the Quality of English Language

I would appreciate more formal language and presentation of material

Round 2

Reviewer 3 Report

Comments and Suggestions for Authors

Dear authors and editors,

happy holidays to you!

The authors did a lot of work on the article in terms of presenting their original experimental findings. The article is ready for publication, a couple of small technical corrections: line 263 - check +, add a space; line 391 - check space and line 395 - silicate(?). It’s not very clear what you mean, impurity in silicate?

This articles will be a nice contribution to Minerals. I highly appreciate the originality of obtained results and authors thorough work on the reviewers comments and suppose that it will be cited.

It is not my area of expertise, but I would like to suggest a wavier (discount) for that contribution, if appliciable.
